# Alleviation of Lipid Disorder and Liver Damage in High-Fat Diet-Induced Obese Mice by Selenium-Enriched *Cardamine violifolia* with Cadmium Accumulation

**DOI:** 10.3390/nu16183208

**Published:** 2024-09-22

**Authors:** Junying Zhu, Qingqing Lv, Fengna Li, Ping Xu, Ziyu Han, Aolin Yang, Zhan Shi, Chao Wang, Jie Jiang, Yunfen Zhu, Xiaofei Chen, Lvhui Sun, Xin Gen Lei, Ji-Chang Zhou

**Affiliations:** 1School of Public Health (Shenzhen), Shenzhen Campus of Sun Yat-sen University, Shenzhen 518107, China; zhujy85@mail2.sysu.edu.cn (J.Z.); lvqq6@mail2.sysu.edu.cn (Q.L.); lifengn@mail2.sysu.edu.cn (F.L.); xuping8@mail2.sysu.edu.cn (P.X.); hanzy8@mail2.sysu.edu.cn (Z.H.); yangaolin@mail2.sysu.edu.cn (A.Y.); shizh28@mail2.sysu.edu.cn (Z.S.); 2Shenzhen Center for Disease Control and Prevention, Shenzhen 518055, China; raulw2003@163.com (C.W.); szjiangjie@139.com (J.J.); 3Enshi Autonomous Prefecture Academy of Agricultural Sciences, Enshi 445000, China; azhu19@163.com (Y.Z.); chenxiaofei_05@163.com (X.C.); 4Hubei Hongshan Laboratory, College of Animal Science and Technology, Huazhong Agricultural University, Wuhan 430070, China; lvhuisun@mail.hzau.edu.cn; 5Department of Animal Science, Cornell University, Ithaca, NY 14853, USA; 6Guangdong Province Engineering Laboratory for Nutrition Translation, Shenzhen 518107, China; 7Guangdong Provincial Key Laboratory of Food, Nutrition and Health, Guangzhou 510080, China

**Keywords:** obesity, metabolic disorder, selenoprotein, heavy metal

## Abstract

Background/Objectives: As a hyperaccumulator of selenium (Se), *Cardamine violifolia* (*Cv*) and its peptide extract could ameliorate the negative effects of a high-fat diet (HFD). However, the effects of the coaccumulation of cadmium (Cd) in Se-enriched *Cv* (*Cv2*) and the potential confounding effect on the roles of enriched Se remain unknown. We aimed to investigate whether *Cv2* could alleviate HFD-induced lipid disorder and liver damage. Methods: Three groups of 31-week-old female mice were fed for 41 weeks (*n* = 10–12) with a control *Cv*-supplemented diet (*Cv1*D, 0.15 mg Se/kg, 30 µg Cd/kg, and 10% fat calories), a control *Cv*-supplemented HFD (*Cv1*HFD, 45% fat calories), and a *Cv2*-supplemented HFD (*Cv2*HFD, 1.5 mg Se/kg, 0.29 mg Cd/kg, and 45% fat calories). Liver and serum were collected to determine the element concentrations, markers of liver injury and lipid disorder, and mRNA and/or protein expression of lipid metabolism factors, heavy metal detoxification factors, and selenoproteins. Results: Both *Cv1*HFD and *Cv2*HFD induced obesity, and *Cv2*HFD downregulated Selenoi and upregulated Dio3 compared with *Cv1*D. When comparing *Cv2*HFD against *Cv1*HFD, *Cv2* increased the liver Se and Cd, the protein abundance of Selenoh, and the mRNA abundance of 10 selenoproteins; reduced the serum TG, TC, and AST; reduced the liver TG, lipid droplets, malondialdehyde, and mRNA abundance of *Mtf1* and *Mt2*; and differentially regulated the mRNA levels of lipid metabolism factors. Conclusions: *Cv2* alleviated HFD-induced lipid dysregulation and liver damage, which was probably associated with its unique Se speciation. However, further research is needed to explore the interaction of plant-coenriched Se and Cd and its effects on health.

## 1. Introduction

As one of the trace essential elements for both humans and animals, selenium (Se) serves as a critical and unique constitutive part of selenocysteine (Sec) in 25 human and 24 rodent selenoproteins [1]. Both organic and inorganic forms of Se from external sources can be efficiently absorbed via the gastrointestinal tract. Organic forms of Se generally exhibit lower toxicity, higher bioavailability, and better antioxidant properties than inorganic forms [2]. Plants can serve as carriers of beneficial elements for the maintenance of body health and are major sources of Se supplementation, as they are capable of biotransforming selenate and selenite into organic Se forms [3].

*Cardamine violifolia* (*Cv*) is a medicinal and edible plant, characterized by being a hyperaccumulator of Se [4]. Moreover, by enriching Se in the major forms of selenocystine and methylselenocystine [5], Se-enriched *Cv* is being developed as a promising Se-rich source. Concerningly, soils in Se-rich areas in the Enshi region of China often contain high levels of cadmium (Cd) [6,7]. In addition to Se, *Cv* could also accumulate the harmful heavy metal Cd [8], and the accumulation of Cd by *Cv* increases with the level of Se fertilization [8,9], thus yielding Se-enriched *Cv* (*Cv2*). The most common type of Cd exposure in non-smoking populations occurs through food and water contaminated with Cd [10], and Cd continuously accumulates in tissue due to the absence of effective excretory mechanisms. As exemplified by the severe Cd poisoning-induced itai-itai disease [11], the long-term accumulation of Cd can eventually cause damage to the functions of various organs in the body, including the liver, kidneys, bones, testes, and others [12]. Interestingly, recent studies have shown that *Cv2*-derived endogenous Se has the potential to antagonize Cd-induced renal damage, skeletal metabolism disorders, and hepatotoxicity [8,9]. Furthermore, reports of Cd-related toxicity are uncommon in areas with high geologic backgrounds of Se and Cd, which may be attributed to the interaction between Se and Cd in plants, leading to alterations in the forms of Cd present in plants that enter the human body, thereby reducing the toxicity of Cd [13,14]. On the other hand, Se can partially alleviate the body health damage caused by Cd exposure [14,15].

It is important to note that obesity has also progressively become one of the most important public health problems in the world [16] and is a significant risk factor for the development and progression of multiple diseases, such as type 2 diabetes, hypertension, hyperlipidemia, non-alcoholic fatty liver disease, and immune disorders [17,18,19]. Many animal studies have found that Se plays a beneficial role in improving lipid metabolic disorder and liver damage [20,21,22], and *Cv*-derived Se-enriched peptides (CSP) could alleviate those phenotypes induced by a high-fat diet (HFD) [23]. However, combined exposure to a HFD and Cd often results in stronger toxic effects on the body [24,25,26], while the effect of Se in the presence of plant-derived Cd accumulation on the adverse health consequences induced by a HFD in Se–Cd-rich areas remains unknown.

Thus, by applying *Cv2* to feed adult female mice with varied dietary fat for more than 40 weeks, our study was to investigate the effect of *Cv2*-derived endogenous Se, in the presence of Cd accumulation, on the lipid metabolic disorders and liver injuries caused by the long-term exposure to HFD. Additionally, we hope to provide some new insights into the effects of Se on ameliorating adverse effects of HFD and Cd caused by consuming foods in naturally Se–Cd-rich areas.

## 2. Materials and Methods

### 2.1. Diets and Animals

The control *Cv* (*Cv1*) and *Cv2* powders used in this study were provided by the Enshi Academy of Agricultural Sciences. Regarding the cultivation methods and conditions of *Cv*, briefly, *Cv1* was cultivated in soil with Se and Cd concentrations of 1.22 mg/kg and 0.61 mg/kg, respectively. On a field adjacent to that for *Cv1* planting, during the same season, *Cv2* was obtained by cultivating the same strain of *Cv1* with Se fertilizer-treated soil (Enshi Zhonghui Selenium Enriched Agricultural Technology Development Co., Ltd., Enshi, China) at a concentration of ≥200 mg Se/kg and with sodium selenite (purity of 97%) applied through foliar spraying. After the two types of whole plants were harvested, they were rinsed, dried, ground, and sieved to obtain a *Cv1* powder (total Se: 7.24 mg/kg; total Cd: 1.4 mg/kg) and a *Cv2* powder (total Se: 76.95 mg/kg; total Cd: 14.5 mg/kg). Over 80% of the Se in these powders existed in the forms of selenocystine and methylselenocystine [5].

The three types of feed used during the experiment were prepared by the Trophic Animal Feed High-Tech Co., Ltd., Nantong, China. For the Se-deficient diet used in all three types of feed, except for Se, all components of this diet met the standards of AIN93. The *Cv1* powder was supplemented into the Se-deficient diet to contain 0.15 mg Se/kg and 0.03 mg Cd/kg, with fat providing 10% or 45% of the energy, to obtain the *Cv1*-supplemented control diet (*Cv1*D) or the *Cv1*-supplemented HFD (*Cv1*HFD), respectively. The *Cv2* powder was used to replace the same amount of *Cv1* powder in *Cv1*HFD to obtain the *Cv2*-supplemented HFD (*Cv2*HFD), containing 1.5 mg Se/kg and 0.29 mg Cd/kg.

All animal experiments reported in this manuscript were approved by the Animal Care and Use Committee of the Institute of Sun Yat-sen University and adhered to the criteria outlined in the ‘Guide of the Care and Use of Laboratory Animals’. Thirty-four female C57BL/6J mice aged 31 weeks, purchased from the Guangdong Medical Laboratory Animal Center, were randomly divided into three groups and fed *Cv1*D (*n* = 12), *Cv1*HFD (*n* = 12), and *Cv2*HFD (*n* = 10) based on the baseline body weight (BW). The animals were maintained under standard conditions, including 4–5 mice per cage, a temperature of 25 ± 2 °C, 70–75% humidity, and a light/dark cycle of 12 h, with free access to food and water during the experimentation. The experimental period was 41 weeks, and the BW was monitored every four weeks for each mouse. At the end of the experiment, all mice were fasted overnight and anesthetized with sodium pentobarbital to minimize pain before being euthanized by heart bleeding.

### 2.2. Oral Glucose Tolerance Test (OGTT) and Insulin Tolerance Test (ITT)

At the 40th week of the animal experiment, all mice were fasted for 6 h. Tail vein blood glucose was measured with a glucometer (ACCU-CHEK Performa, Roche Diagnostics GmbH, Mannheim, Germany). The OGTT was performed using an oral gavage of 2 g/kg BW of glucose, and blood glucose was measured at 0, 15, 30, 60, and 120 min after glucose loading. In the middle of the 41st week of the experiment, all mice were fasted for 4 h. Then, the ITT was performed using an intraperitoneal injection of 1 U/kg BW of insulin, and blood glucose was measured at 0, 15, 30, 60, 90, and 120 min after insulin loading with a glucometer. Prior to measuring the tail vein blood glucose and administering insulin via intraperitoneal injection, mild and non-irritating disinfection with an iodine tincture was performed on the local area to minimize animals’ pain and stress and to reduce the risk of infection.

### 2.3. Sample Preparation and Biochemical Analysis

Blood and tissue samples were collected promptly for analysis or cryopreservation. The serum samples were isolated by centrifugation at 12,000× *g* for 10 min at 4 °C. Approximately 50 mg liver samples were homogenized and centrifugated at 2500× *g* for 10 min at 4 °C to collect the supernatant. For the prepared serum and liver homogenate samples, commercial kits were used to measure the concentrations of TG (Fujifilm Wako Pure Chemical Co., Osaka, Japan), total cholesterol (TC), and low-density lipoprotein cholesterol (LDL-C) (Nanjing Jiancheng Bioengineering Institute, Nanjing, China). The serum activity of alanine aminotransferase (ALT) and aspartate transaminase (AST) was measured by a HITACHI Automatic Analyzer 3100 at the Laboratory Animal Center of Sun Yat-sen University.

### 2.4. Elemental Analysis

Element quantification in the *Cv1*, *Cv2*, *Cv1*D, *Cv1*HFD, *Cv2*HFD, and liver tissue was performed using an Agilent 7850 Inductively Coupled Plasma Mass Spectrometer (Agilent Technologies, Santa Clara, CA, USA). Briefly, for sample digestion using an oven, 5 mL of nitric acid was added to a fluoroplastic liner containing an individual sample, and the samples in the insert were pre-incubated for 30 min at 80 °C. Subsequently, they were incubated for 2 h at 110 °C, followed by an additional incubation step for 4 h at 160 °C. Then, the dissolved samples were allowed to cool down to room temperature and quantitatively transferred to test tubes. Finally, a specific volume of deionized water was added to the test tubes, which were then vortexed and further assayed.

### 2.5. Malondialdehyde (MDA) Levels and Total Antioxidant Capacity (T-AOC)

To analyze the MDA levels, 20 mg liver samples were homogenized with phosphate-buffered saline using a tissue grinder. The homogenate was centrifuged at 3000× *g* for 10 min at 4 °C, and the supernatant was collected for analysis using an MDA assay kit (Nanjing Jiancheng Bioengineering Institute, Nanjing, China). For T-AOC measurement, the liver homogenate was centrifuged at 12,000× *g* for 5 min at 4 °C. Then, the supernatant was collected for analysis using a T-AOC assay kit (Nanjing Jiancheng Bioengineering Institute, Nanjing, China).

### 2.6. Histological Analysis

Six mice were randomly selected from each group and the histopathology of hepatic steatosis was evaluated by determining the average number, size, and area of hepatic lipid droplets. The freshly isolated livers of the mice were fixed in 10% neutral buffered formalin and then embedded in paraffin and sliced at 5 μm. After this, sections of the liver were stained with hematoxylin and eosin (H&E). The quantitative analysis of the number and size of the lipid droplets was performed using the ImageJ software 1.52v with three fields per slide, and the mean values were calculated.

### 2.7. RNA Extraction and Quantitative Real-Time PCR (qPCR)

Total RNA was extracted from the liver tissue using a total RNA extraction reagent (Yeasen Biotech (Shanghai) Co., Ltd., Shanghai, China), according to the manufacturer’s instructions. The concentration and purity of the RNA were determined by measuring the absorbance at 260/280 nm using a spectrophotometer (NanoDrop, Thermo Fisher Scientific, Waltham, MA, USA). The RNA integrity was checked by agarose gel electrophoresis. cDNA was synthesized using a reverse transcription kit (Takara Biomedical Technology (Beijing) Co., Ltd., Beijing, China), and then qPCR was performed. All primers for the selenoprotein genes and lipid metabolism-related genes used in the qPCR are listed in Appendix A. Relative quantification was performed using the 2^−ΔΔCT^ method normalized to the control *Actb*.

### 2.8. Western Blot Assay

The Western blotting experiment was performed according to a protocol previously described [27]. Briefly, 20 mg liver samples were homogenized in ice-cold RIPA lysis buffer (Beyotime Biotechnology, Shanghai, China) containing complete protease inhibitor using a tissue grinder. The homogenate was centrifuged at 12,000× *g* for 15 min at 4 °C, and the supernatant was collected. The protein concentration was determined using the BCA Protein Assay Kit (Thermo Fisher Scientific). After this, equal amounts of protein (usually 20 μg) from each sample were separated by SDS-PAGE and transferred onto a PVDF membrane (0.22 or 0.45 μm, Millipore, Burlington, MA, USA). Then, the membrane was blocked with 5% non-fat milk in a Tris–Tween-buffered saline buffer (TBST) for 2 h and was incubated with primary antibodies (Appendix A) overnight at 4 °C. After washing with TBST, the membrane was incubated with horseradish peroxidase-conjugated secondary antibodies (dilution: 1:5000) for 1 h at room temperature. Finally, blots were developed on the membrane using MiniChemiTM910 reagents (Beijing SinSage Technology Co., Ltd., Beijing, China). We digitally quantified the resultant signals and normalized the data to the Actb abundance.

### 2.9. Statistical Analysis

All data were presented as the mean ± SEM. Statistical analysis was performed using GraphPad Prism 9.51 (GraphPad Software Inc., San Diego, CA, USA) and SPSS 26.0 (SPSS Inc., Chicago, IL, USA). The differences between the groups were determined using one-way or two-way ANOVA with Tukey’s multiple comparison test. The results were considered statistically significant at *p* < 0.05.

## 3. Results

### 3.1. Growth Performance

Over the 41 weeks of HFD administration, we found that *Cv2* did not significantly alleviate HFD-induced weight gain in female mice. As shown in Figure 1, there was no significant difference in BW between the *Cv1*HFD and *Cv2*HFD groups (*p* ≥ 0.05).

### 3.2. Concentrations of Trace Elements in the Liver

Compared with those of the *Cv1*D or *Cv1*HFD group, the accumulation of Se significantly increased by ~22% in the livers of the *Cv2*HFD group (*p* < 0.0001; Figure 2A), while the accumulation of Cd significantly increased by 360% or 500%, respectively (*p* < 0.0001; Figure 2B). However, there was no difference in either Se or Cd between the *Cv1*D and *Cv1*HFD groups (*p* ≥ 0.05). We also observed a significant association between the Se and Cd levels in the *Cv1*D group (*R*^2^ = 0.4822, *p* = 0.0379) and all mice (*R*^2^ = 0.5612, *p* < 0.0001; Figure 2C). Furthermore, no differences were observed in the concentrations of copper, iron, and zinc among the *Cv1*D, *Cv1*HFD, and *Cv2*HFD groups (*p* ≥ 0.05; Appendix A).

### 3.3. Glucose Metabolism

After the mice were fasted for 12 h before the OGTT test, the blood glucose levels in the *Cv1*HFD group and *Cv2*HFD group were both significantly higher than those in the *Cv1*D group at 60 min (*p* < 0.05; Appendix A). Interestingly, although differences were not observed at other time points or in the glucose area under the curve (AUC) of OGTT among the three groups, there was a trend toward lower blood glucose levels in the *Cv2*HFD group compared with the *Cv1*HFD group (Appendix A). For the ITT assessment, the fast plasma glucose level and the blood glucose level at 30 min after the injection of insulin in the mice in the *Cv1*HFD and *Cv2*HFD groups were significantly higher than those in the *Cv1*D group (*p* < 0.05; Appendix A). However, no difference in the glucose AUC was observed among the three groups (*p* ≥ 0.05; Appendix A).

### 3.4. Lipid Disorder and Liver Injury

Compared with the *Cv1*HFD group, *Cv2* significantly decreased the serum levels of TG (20%, *p* < 0.05; Figure 3A) and TC (210%, *p* < 0.05; Figure 3B), as well as liver TG (69%, *p* < 0.05; Figure 3D), in the *Cv2*HFD group, without a significant effect on the serum LDL-C (*p* ≥ 0.05; Figure 3C) and liver TC and LDL-C (*p* ≥ 0.05; Figure 3E,F). Furthermore, although no difference was observed in the serum ALT activity (*p* ≥ 0.05; Figure 3G), *Cv2* alleviated HFD-induced liver injuries, characterized by a ~33% decrease in serum AST activity in the *Cv2*HFD group compared with the *Cv1*HFD group (*p* < 0.05; Figure 3H).

### 3.5. Regulating Gene and Protein Expression in Lipid Metabolism

In the *Cv1*HFD group, unclear intercellular boundaries, shrinking nuclei squeezed to the edges of the cell, and the disordered arrangement of hepatic cords were observed (Figure 4A). In the *Cv2*HFD group, hepatic lipid droplets were significantly fewer and smaller, with clearer boundaries of liver cells than those in the *Cv1*HFD group (*p* < 0.05). However, there was no significant difference in the liver weight among the *Cv1*D, *Cv1*HFD, and *Cv2*HFD groups (*p* ≥ 0.05; Figure 4B).

The mRNA levels of key genes (*Gpat1*, *Gpat2*, *Dgat1*, and *Dgat2*) involved in TG synthesis showed no significant differences in the livers of *Cv1*HFD-fed female mice compared with *Cv1*D-fed female mice (*p* ≥ 0.05), but a gene involved in TG hydrolysis (*Lipc*) was significantly downregulated (*p* < 0.05; Figure 4C). In contrast, in the livers of *Cv2*HFD-fed mice, genes involved in TG synthesis (*Gpat1* and *Dgat2*) and genes involved in TG hydrolysis (*Atgl* and *Hsl*) were significantly downregulated compared to the *Cv1*HFD group, while the mRNA levels of *Lipc* and *Lpl* were significantly upregulated (*p* < 0.05).

Regarding cholesterol homeostasis in the livers of *Cv1*HFD-fed mice, a key gene involved in cholesterol synthesis, *Hmgcr*, was significantly downregulated compared to *Cv1*D-fed mice, and a gene involved in cholesterol transport, *Abcg1*, was upregulated (*p* < 0.05; Figure 4D). In the livers of *Cv2*HFD-fed mice, the mRNA levels of *Hmgcr*, *Sqle*, and *Ldlr* were significantly upregulated compared to the *Cv1*HFD and/or *Cv1*D group(s), while the mRNA levels of *Abca1*, *Abcg1*, and *Abcg8* were significantly downregulated (*p* < 0.05).

### 3.6. Hepatic Oxidative Stress and Antioxidant Defense Capacity

Compared with those of the *Cv1*HFD group, *Cv2* significantly downregulated the elevated mRNA expression levels of metal regulatory transcription factor 1 (*Mtf1*) and metallothionein 2 (*Mt2*) caused by the HFD in the livers of the *Cv2*HFD group (*p* < 0.05; Figure 5A,C). Furthermore, no differences in the relative protein levels of Mtf1, Mt, and Sod1, as well as the T-AOC level, were observed among the *Cv1*D, *Cv1*HFD, and *Cv2*HFD groups (*p* ≥ 0.05; Figure 5E–H). However, we found that, compared with the *Cv1*HFD group, *Cv2* significantly decreased the liver MDA level in the *Cv2*HFD group by ~75% (*p* < 0.05; Figure 5I).

### 3.7. Relative mRNA and Protein Expression of Selenoproteins

The mRNA abundance of all selected selenoprotein genes showed no differences (*p* ≥ 0.05) between the *Cv1*D and *Cv1*HFD groups (Figure 6A,B; Appendix A). However, compared with those in the *Cv1*D and/or *Cv1*HFD group(s), the expression of 12 selenoprotein genes (*Dio1*, *Dio3*, *Gpx4*, *Msrb1*, *Selenof*, *Selenoo*, *Selenos*, *Selenow*, *Sephs2*, *Txnrd1*, *Txnrd2*, and *Txnrd3*) was significantly upregulated, while *Selenoi* was significantly downregulated in the livers of the *Cv2*HFD group (*p* < 0.05; Figure 6A,B).

Compared with those of the *Cv1*D group, the relative protein levels of the ER-resident Selenoi and the non-ER-resident Dio3 were, respectively, downregulated and upregulated in the livers of the *Cv2*HFD group (*p* < 0.05; Figure 6C–F). Furthermore, compared with those of the *Cv1*HFD group, the relative protein level of Selenoh was upregulated in the livers of the *Cv2*HFD group (*p* < 0.05; Appendix A). Although there were no differences in the relative protein levels of Selenoo and Txnrd2 among the *Cv1*D, *Cv1*HFD, and *Cv2*HFD groups, there was an upregulatory trend in the *Cv2*HFD group (*p* = 0.05).

## 4. Discussion

The potential of Se-rich plants as a source for the development of novel Se supplements has drawn attention from many researchers due to their ability to improve health [23,28,29]. *Cv*, as a hyperaccumulator of Se, serves as an ideal plant-derived Se source. Previous studies have found that CSP from *Cv* may alleviate HFD-induced disorder of lipid metabolism through potential mechanisms such as increased thermogenesis and reduced oxidative stress and inflammation, as well as the regulation of gene expression in lipid and cholesterol metabolism [23]. However, *Cv* is also able to hyperaccumulate Cd, which hinders its promotion as an ideal Se supplement and calls for further investigations of the health effects of Se–Cd coenrichment in this species, as well as other plants [14,15]. Zhang et al. [8,9] aquacultured three types of Se–Cd-coenriched *Cv*s by supplementing 0, 1, or 2 mg/L of sodium selenate and 35 mg/L of cadmium chloride (CdCl_2_) in the culturing solution and found that both Se (0.2–0.86 g/kg) and Cd (4.2–5.4 g/kg) in *Cv* positively increased with the Se supplementation. By gavage in mice with a purified water control or the same dosage of Cd (3.2 mg/kg BW) in the form of CdCl_2_ or any of the three Se–Cd-coenriched *Cv*s, the Cd-induced renal damage, skeletal metabolism disorder [9], and hepatotoxicity [8] were improved with the increased Se levels across the three Se–Cd-coenriched *Cv*s, in which selenocystine probably played a critical role.

Given the characteristic of *Cv* being able to accumulate Se and Cd, we observed the naturally increased enrichment of Cd from the soil with Se supplementation in our study, which was consistent with the findings from other studies [7,8]. In our studied animals, the Cd exposure levels were much lower than those reported to cause significant liver damage [8,9,24,30,31,32,33]. Additionally, in the intervention model, the Se/Cd ratio (1.5 mg/kg Se vs. 0.29 mg/kg Cd) in our study was 5.17, while, in other studies showing the protective effect of Se against Cd exposure, the ratios were 0.16 (0.5 mg/kg Se vs. 3.2 mg/kg Cd) [8,9] and 0.8 (1.6 mg/kg Se vs. 2 mg/kg Cd) [34], much lower than ours. Se supplementation has been shown to improve the adverse health effects induced by HFD or Cd exposure; however, combined exposure to Cd and HFD often exhibits a synergistic effect in terms of damage to the health of the body [24,33,35]. Meanwhile, the role of Se in the presence of Cd accumulation on long-term HFD-induced health outcomes remains unknown. Thus, we conducted this study to investigate the effects of *Cv2* on metabolic disorder and liver damage induced by the long-term consumption of a HFD and to explore the role of Se in the plant-derived high background of Cd.

We found that, despite the tenfold increase in both the Se and Cd levels in *Cv2*HFD compared to *Cv1*D and *Cv1*HFD, there was a significant difference in the accumulation levels of Se and Cd in the livers. Indeed, under physiological conditions, there is a mechanism in the body to maintain Se homeostasis, primarily regulated through urinary excretion rather than absorption [36]. However, mammals lack effective regulatory mechanisms to control the balance of Cd levels in the body. Thus, the small amount of Cd entering the body through food or other pathways could be excreted in minute quantities through the urine, with the majority continuously accumulating in organs like the liver and eventually causing harm to the body [37]. Previous studies have revealed that combined exposure to a HFD and Cd could significantly disrupt the essential metal homeostasis in female mice, which could be associated with influencing the expression of metal transporters [24]. However, this phenomenon was not observed in our study. This could be explained by Se being capable of alleviating the effect of the HFD and Cd in terms of disrupting essential metal homeostasis by regulating metal transporters to some extent [24,38]. It is well known that a long-term HFD can disrupt lipid metabolism, leading to obesity, lipid metabolic disorder, and liver damage [39,40]. Furthermore, chronic low Cd exposure (150 μg/L CdCl_2_ or 92 μg/L Cd spiked in pure water) at food limitation-relevant levels is sufficient to cause significantly disrupted hepatic energy metabolism homeostasis [41]. Therefore, the presence of Cd in *Cv2* may influence the protective effect of Se against the metabolic disorder induced by a long-term HFD, potentially exacerbating the health damage in synergy with the HFD. Although we found that *Cv2* did not significantly influence glucose tolerance or insulin sensitivity in HFD-induced female obese mice, we observed that the supplementation of *Cv2* could alleviate the increase in serum and hepatic lipid levels, as well as liver injury marker levels, induced by the HFD in mice, which may be attributed to the protective effects of *Cv*-derived endogenous Se on lipid metabolic disorder and liver injury induced by a HFD and Cd [23,29,42].

Lipid disorder is closely associated with abnormalities in lipid (including TG and TC) synthesis, transport, and metabolism. Previous studies have found that Cd can downregulate the expression of Atgl and Lpl [43], while, in HFD-fed mice, CSP increased the expression of Lpl and reduced the expression of Hmgcr [23]. Similarly, another study also demonstrated that, in normal diet-fed broilers, Se-enriched *Cv* could upregulate the expression of LPL and reduce the expression of HMGCR in the liver, without an effect on DGAT2. Obviously, Se, Cd, and HFD could affect lipid homeostasis by influencing the expression of lipid metabolism-related factors. In our study, *Cv2* downregulated the gene expression of TG synthesis enzymes (*Gpat1* and *Dgat2*) and upregulated TG hydrolytic enzymes (*Lipc* and *Lpl*), which are helpful in alleviating the HFD-induced elevation of TG levels [44]. Interestingly, some results in our study differed from previous research. We observed that the expression of the other two TG hydrolytic enzyme genes (*Atgl* and *Hsl*) was downregulated by *Cv2* supplementation in HFD-fed mice. Furthermore, we observed that *Cv2* upregulated *Hmgcr* in the *Cv2*HFD group compared with the *Cv1*HFD group. These findings, differing from previous research, suggest that the plant-derived Cd accumulation in the mouse liver affected Se’s antagonistic effect on the HFD. We also observed that *Cv2* reduced the expression of cholesterol transport-related genes (*Abca1*, *Abcg1*) while increasing the expression levels of *ldlr* involved in cholesterol clearance in the *Cv2*HFD group, which are helpful in reducing cholesterol efflux and promoting its clearance [45]. This might have contributed to the lower serum TC level in the *Cv2*HFD group than in the *Cv1*HFD group [45,46,47]. Therefore, although its effectiveness was attenuated by the presence of plant-derived Cd, the endogenous Se in *Cv2* still demonstrated beneficial improvements in lipid metabolic disorder in HFD-induced obese mice.

Mts, particularly Mt1 and Mt2, are widely present in the parenchymal organs, such as the liver and kidneys, of mice, while Mtf1 serves as a positive regulatory factor for the expression of *Mts* [48,49]. They mainly participate in the detoxification of heavy metals, free radical scavenging, and protection against oxidative damage and are capable of inhibiting the toxicity of Cd [49]. Consistent with previous studies, we observed the upregulation of the gene expression levels of *Mtf1*, *Mt1*, and *Mt2* in the mouse liver induced by *Cv1*HFD; this may be a response of the body to HFD-induced oxidative stress [50,51]. Furthermore, we found that *Cv2* supplementation reduced the expression of these genes. Previous studies have found that exposure to Cd could upregulate the expression of *Mtf1*, *Mt1*, and *Mt2*, while Se reversed this effect [52], and the observed phenomenon between the *Cv1*HFD and *Cv2*HFD groups in our study may be attributed to the antagonistic effects of Se against Cd and the HFD. However, we observed no difference in the relative protein levels of Mt and Mtf1 among the three groups. In fact, a study also found that a HFD significantly increased the mRNA expression of *Mt2*, and the protein expression levels of Mts showed no significant difference compared to the control group [24], while another study reported the opposite results [33]. In fact, Mts primarily participate in maintaining the homeostasis of essential trace elements such as copper and zinc in the body [53]. Cd exposure stimulates the synthesis of hepatic Mts, which bind with Cd^2+^ to form Cd–MT complexes to reduce the damage caused by free Cd^2+^ [53]. Previous studies have indicated that combined exposure to Se and Cd results in the formation of Cd–Se complexes or Cd–Se protein complexes [54,55]. Therefore, the presence of endogenous Se in *Cv2* may have led to the formation of inert complexes with Cd^2+^, reducing the need for Mts in the organism.

Furthermore, selenoproteins partly participated in alleviating the negative effects of the HFD and Cd exposure. SELENOP is essential in maintaining the homeostasis of selenoproteins and is also associated with lipid metabolism [56]. Elevated circulating levels of SELENOP have been observed in obese individuals [57], and a deficiency in Selenof [58], Selenom [59], Selenos [60], Selenot [61], or Selenov [62] has been shown to lead to lipid metabolic disorder in established mouse models and usually exacerbates HFD-induced lipid accumulation in the body. At the same dietary Se level, the *Cv1*D and *Cv1*HFD groups had no differences in the mRNA abundance of the selenogenome, regardless of the dietary fat content. The supplementation of *Cv2* not only significantly increased the Se concentrations in the livers of the *Cv2*HFD group but also differentially regulated the mRNA expression levels of some selenoproteins. Previous studies have indicated that selenoproteins, particularly some endoplasmic reticulum-resident selenoproteins, play crucial roles in mediating Se’s antagonism against Cd-induced hepatic toxicity [63]. Interestingly, we observed that the protein abundance of some selenoproteins in the liver was not consistent with their corresponding mRNA expression. This may be caused by Cd in *Cv2* inhibiting the biosynthesis of selenoproteins and Se metabolites to weaken the beneficial effects of Se on the organism [63]. In addition, the suppression of selenoprotein synthesis by Cd may also be due to the formation of the above-mentioned Cd–Se complex [63,64]. Although the specific selenoproteins involved and their respective abilities to bind Cd are not yet clear, the significantly changed protein levels of Dio3, Selenoh, and Selenoi in the *Cv2*HFD group suggest that these selenoproteins may have specific roles in mediating Se’s effect in the interaction of the HFD and Cd.

Lastly, it should be noted that our study has certain limitations: although the Se content was higher and the Cd content was lower in *Cv2*HFD in our study than in previous studies [8,9], we were unable to evaluate in detail the impact of the interaction between Se and Cd on HFD-induced health damage due to a shortage of *Cv* with high Se as in *Cv2*, low Cd as in *Cv1*, low Se as in *Cv1*, or high Cd as in *Cv2* without exogenous element supplementation. Furthermore, previous studies have reported that, in Se–Cd-coenriched plants, Se could alter the forms and toxicity of Cd to the body [13,14,64]. However, we did not identify the forms of Cd present in *Cv2* or their different impacts on health. Therefore, future studies need to illuminate the interaction between Se and Cd in the soil–plant–organism system in Se–Cd-rich areas. It would also be worthwhile to further investigate the utilization and safety of Se, changes in Cd toxicity in Se–Cd-coenriched agricultural products, and the impact of the interaction between Se, Cd, and the HFD on overall health.

## 5. Conclusions

In summary, our study suggests that, in the presence of Cd accumulation, *Cv2* still demonstrates effects in terms of alleviating the elevated levels of serum TC and TG, as well as liver lipid deposition, histological damage, and oxidative stress, in HFD mice. The observed beneficial effects of *Cv2* on HFD-induced metabolic disorder were probably associated with its unique Se speciation to antagonize the combined exposure to the HFD and Cd via its regulation of the expression of specific lipid metabolic genes and selenoproteins. However, the current study suggests that, when considering the consumption of Se-enriched agricultural products as a means of Se supplementation, it is necessary to evaluate the potential toxicity caused by the coaccumulation of Cd. Further research is still needed to explore the interaction of Se, Cd, and the HFD and its effects on health to provide a better reference for the development of Se supplements and Se-enriched agricultural products originating from naturally Se–Cd-rich areas.

## Figures and Tables

**Figure 1 nutrients-16-03208-f001:**
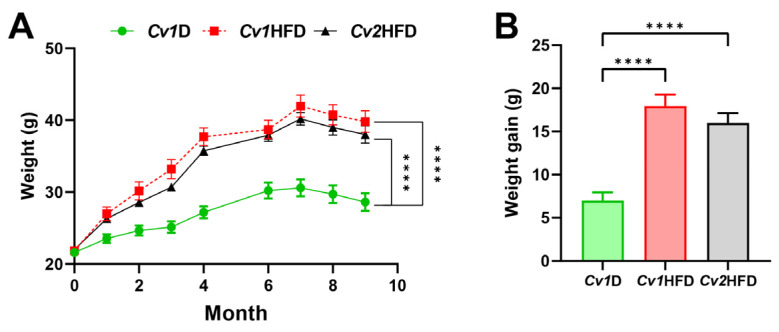
Growth performance in adult female mice during the 41-week experiment. (**A**) Body weight. (**B**) Total weight gain. Data are means ± SEM (*n* = 10–12) and differ with ****, *p* < 0.0001. *Cv1*D, control diet with 0.15 mg Se and 0.03 mg Cd (in the form of control *Cv*)/kg; *Cv*, *Cardamine violifolia*; *Cv1*HFD, high-fat diet with 0.15 mg Se and 0.03 mg Cd (in the form of control *Cv*)/kg; *Cv2*HFD, high-fat diet with 1.5 mg Se and 0.29 mg Cd (in the form of Se-enriched *Cv*)/kg.

**Figure 2 nutrients-16-03208-f002:**
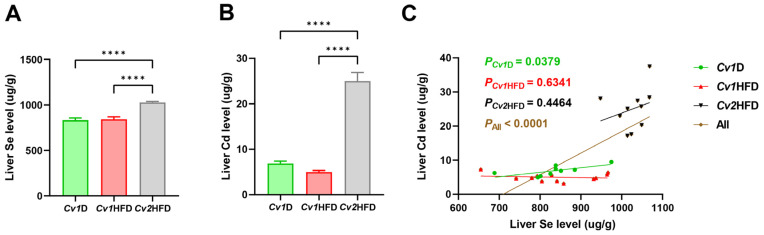
Accumulation of selenium (Se) (**A**) and cadmium (Cd) (**B**) and their correlation (**C**) in the livers of adult female mice. Data are means ± SEM (*n* = 9–12) and differ with ****, *p* < 0.0001. *Cv1*D, control diet with 0.15 mg Se and 0.03 mg Cd (in the form of control *Cv*)/kg; *Cv*, *Cardamine violifolia*; *Cv1*HFD, high-fat diet with 0.15 mg Se and 0.03 mg Cd (in the form of control *Cv*)/kg; *Cv2*HFD, high-fat diet with 1.5 mg Se and 0.29 mg Cd (in the form of Se-enriched *Cv*)/kg.

**Figure 3 nutrients-16-03208-f003:**
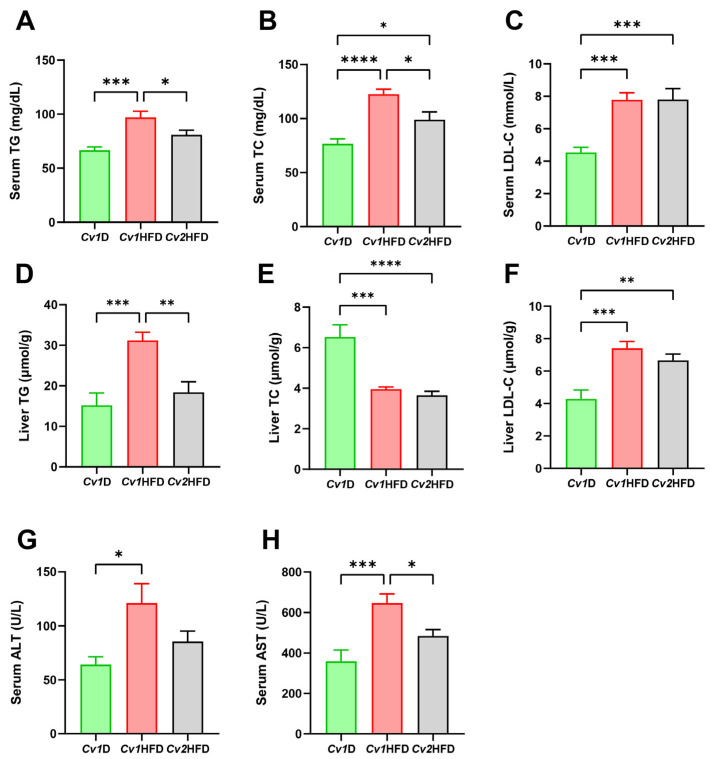
Lipid levels in the serum and liver and hepatic injury in adult female mice. (**A**–**C**) Serum concentrations of TG, TC, and LDL-C. (**D**–**F**) Liver concentrations of TG, TC, and LDL-C. (**G**,**H**) Serum ALT and AST activity. Data are means ± SEM (*n* = 8–10) and differ with *, *p* < 0.05; **, *p* < 0.01; ***, *p* < 0.001; and ****, *p* < 0.0001. ALT, alanine aminotransferase; AST, aspartate transaminase; *Cv1*D, control diet with 0.15 mg Se and 0.03 mg Cd (in the form of control *Cv*)/kg; *Cv*, *Cardamine violifolia*; *Cv1*HFD, high-fat diet with 0.15 mg Se and 0.03 mg Cd (in the form of control *Cv*)/kg; *Cv2*HFD, high-fat diet with 1.5 mg Se and 0.29 mg Cd (in the form of Se-enriched *Cv*)/kg; LDL-C, low-density lipoprotein cholesterol; Se, selenium; TC, total cholesterol; TG, triglyceride.

**Figure 4 nutrients-16-03208-f004:**
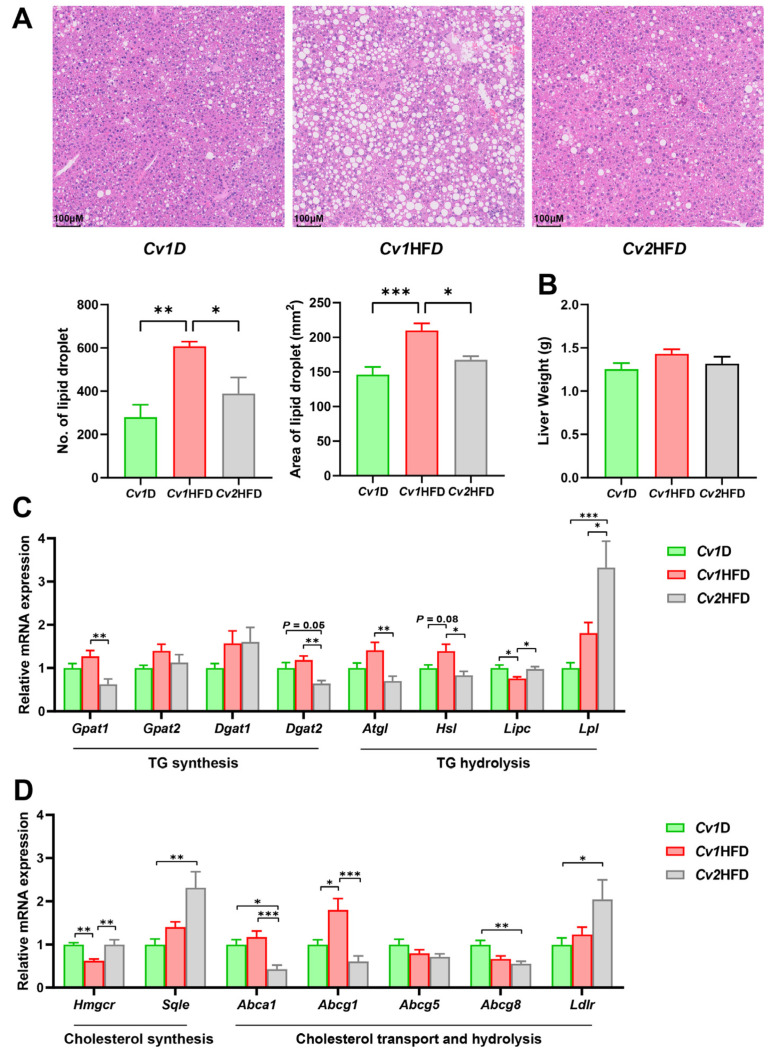
Improvement due to selenium (Se)-enriched *Cardamine violifolia* (*Cv*) in hepatic lipid accumulation and related molecular mechanisms. (**A**) H&E staining of the liver tissue with scale bar = 100 μM; the average number and area of hepatic lipid droplets were quantified with three fields per slide by using the “Analyze particles” function of ImageJ, and the mean values were calculated. (**B**) Total liver weight. (**C**) Expression of metabolic genes of triglycerides. (**D**) Expression of metabolic genes of cholesterol. Data are means ± SEM (*n* = 6–12) and differ with *, *p* < 0.05; **, *p* < 0.01; and ***, *p* < 0.001. *Abca1*, ATP-binding cassette subfamily A member 1; *Abcg1*/*5*/*8*, ATP-binding cassette subfamily G member 1/5/8; *Atgl*, adipose triglyceride lipase; ATP, adenosine triphosphate; *Cv1*D, control diet with 0.15 mg Se and 0.03 mg Cd (in the form of control *Cv*)/kg; *Dgat1*/*2*, diacylglycerol O-acyltransferase 1/2; *Gpat1*/*2*, glycerol-3-phosphate acyltransferase 1/2; *Cv1*HFD, high-fat diet with 0.15 mg Se and 0.03 mg Cd (in the form of control *Cv*)/kg; *Cv2*HFD, high-fat diet with 1.5 mg Se and 0.29 mg Cd (in the form of Se-enriched *Cv*)/kg; *Hmgcr*, 3-hydroxy-3-methylglutaryl coenzyme A reductase; *Hsl*, hormone-sensitive lipase; *Ldlr*, low-density lipoprotein receptor; *Lipc*, lipase C hepatic type; *Lpl*, lipoprotein lipase; No., number; *Sqle*, squalene epoxidase.

**Figure 5 nutrients-16-03208-f005:**
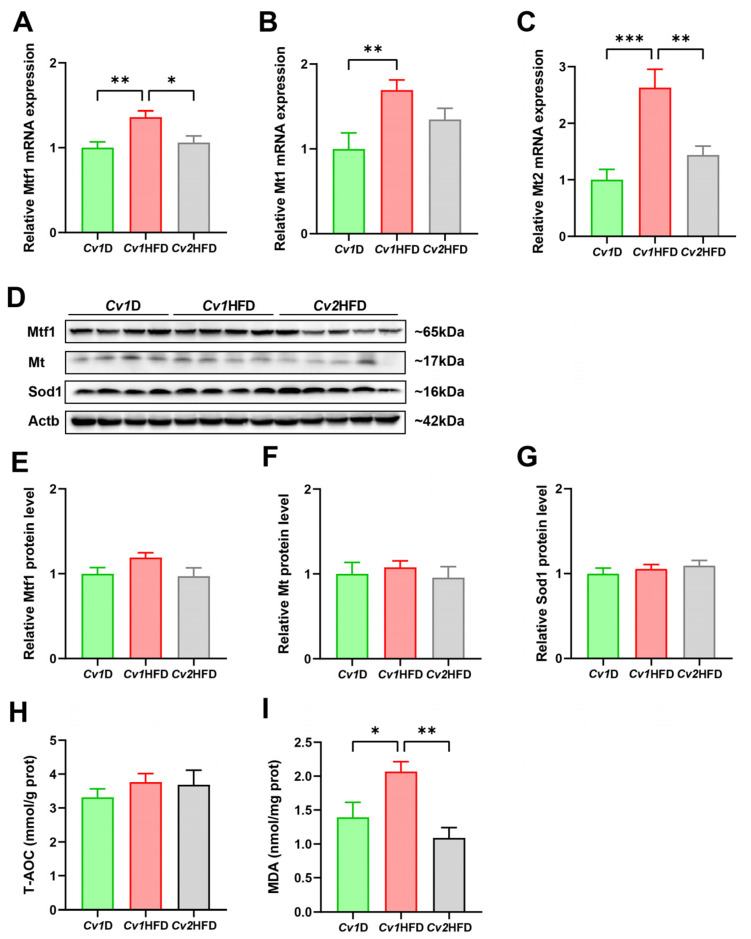
Oxidative stress and antioxidant levels in the livers of adult female mice. (**A**–**C**) The relative mRNA expression of *Mtf1*, *Mt1*, and *Mt2*. (**D**) A representative immunoblot image for Mtf1, Mt, and Sod1. (**E**–**G**) The relative protein levels of Mtf1, Mt, and Sod1. (**H**) The T-AOC level. (**I**) The level of MDA in the liver. Data are means ± SEM (*n* = 8–12) and differ with *, *p* < 0.05; **, *p* < 0.01; and ***, *p* < 0.001. *Cv1*D, control diet with 0.15 mg Se and 0.03 mg Cd (in the form of control *Cv*)/kg; *Cv*, *Cardamine violifolia*; *Cv1*HFD, high-fat diet with 0.15 mg Se and 0.03 mg Cd (in the form of control *Cv*)/kg; *Cv2*HFD, high-fat diet with 1.5 mg Se and 0.29 mg Cd (in the form of Se-enriched *Cv*)/kg; MDA, malondialdehyde; *Mt1*/*2*, metallothionein 1/2; *Mtf1*, metal regulatory transcription factor 1; Se, selenium; Sod1, superoxide dismutase 1; T-AOC, total antioxidant capacity.

**Figure 6 nutrients-16-03208-f006:**
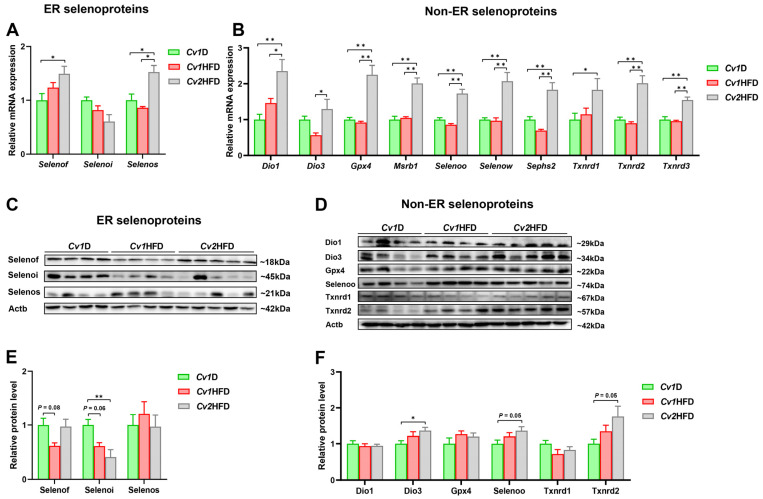
Expression of selenoproteins in the livers of adult female mice. (**A**) The relative mRNA expression levels of *Selenof*, *Selenoi*, and *Selenos*. (**B**) The relative mRNA expression levels of *Dio1*, *Dio3*, *Gpx4*, *Msrb1*, *Selenoo*, *Selenow*, *Sephs2*, *Txnrd1*, *Txnrd2*, and *Txnrd3*. (**C**) A representative immunoblot image for Selenof, Selenoi, and Selenos. (**D**) A representative immunoblot image for Dio1, Dio3, Gpx4, Selenoo, Txnrd1, and Txnrd2. (**E**) The relative protein levels of Selenof, Selenoi, and Selenos. (**F**) The relative protein levels of Dio1, Dio3, Gpx4, Selenoo, Txnrd1, and Txnrd2. Data are means ± SEM (*n* = 8–12) and differ with *, *p* < 0.05 and **, *p* < 0.01. *Cv1*D, control diet with 0.15 mg Se and 0.03 mg Cd (in the form of control *Cv*)/kg; *Cv*, *Cardamine violifolia*; *Dio1/3*, iodothyronine deiodinase 1/3; ER, endoplasmic reticulum; *Gpx4*, glutathione peroxidase 4; *Cv1*HFD, high-fat diet with 0.15 mg Se and 0.03 mg Cd (in the form of control *Cv*)/kg; *Cv2*HFD, high-fat diet with 1.5 mg Se and 0.29 mg Cd (in the form of Se-enriched *Cv*)/kg; *MsrB1*, methionine sulfoxide reductase B1; Se, selenium; *Selenof/i/o/s/w*, selenoprotein F/I/O/S/W; *Sephs2*, selenophosphate synthetase 2; *Txnrd1/2/3*, thioredoxin reductase 1/2/3.

## Data Availability

Data are contained within the article.

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
