# Peer review of "Alleviation of Lipid Disorder and Liver Damage in High-Fat Diet-Induced Obese Mice by Selenium-Enriched Cardamine violifolia with Cadmium Accumulation"

_nutrients, 2024, doi:10.3390/nu16183208_

Round 1

Reviewer 1 Report

Comments and Suggestions for Authors

Alleviation of lipid disorder and liver damage in high-fat diet-induced obese mice by selenium-enriched Cardamine violifolia with cadmium accumulation

Obesity is becoming one of the major prevailing health-care problems worldwide, which is associated with a wide spectrum of metabolic disorders. Excessive fat accumulation and insulin resistance are the two main pathological phenotypes of obesity, which increase the risk of hyperglycemia, dyslipidemia, and hypertension, leading to type 2 diabetes mellitus and cardiovascular diseases.

Study was approved by the Animal 106 Care and Use Committee of the Institute of Sun Yat-sen University, and adhered to the criteria outlined in the ‘Guide of the Care and Use of Laboratory Animals’. Manuscript provided detailed information about the animal process of experimentation;  there is ethical aaprovement; in addition,  the "3R" principles regarding- Replacing animals with alternatives anywhere possible, Reducing the number of animals used, Refinement of experimental conditions and procedures to minimize harm to animals are present. There is no details about animal housing, growth and pain management in this manuscript. Results were presented in 6 figures depicting all dosages made and gene analysis. References must be up-dated. There ar similar article suc as u T, Guo J, Zhu S, Li M, Zhu Z, Cheng S, Wang S, Sun Y, Cong X. Protective effects of selenium-enriched peptides from Cardamine violifolia against high-fat diet induced obesity and its associated metabolic disorders in mice. RSC Adv. 2020 Aug 26;10(52):31411-31424. doi: 10.1039/d0ra04209a. PMID: 35520651; PMCID: PMC9056391.

Author Response

Comment 1: There is no details about animal housing, growth and pain management in this manuscript.

Response 1: Thanks for the reviewer’s comment. We have supplemented detailed information about animal housing, growth, and pain management in Lines 111-117 and Lines 126-128 of the revised manuscript, including “The animals were maintained under standard conditions, including 4-5 mice per cage, a temperature of 25 ± 2 °C, 70–75% humidity, and a light/dark cycle of 12 h, with free access to food and water during the experimentation. The experimental period was 41 weeks, and BW was monitored every four weeks for each mouse. At the end of the experiment, all mice were fasted overnight and anesthetized with sodium pentobarbital to minimize pain before being euthanized by heart bleeding.”; “Prior to measuring tail vein blood glucose and administering insulin via intraperitoneal injection, a mild and non-irritating disinfection with iodine tincture was performed on the local area to minimize animal pain and stress and to reduce the risk of infection.”

Comment 2: References must be up-dated.

Response 2: We appreciate your suggestion, and have updated the references in the revised manuscript to include as many publications from the past five years as possible. Please see the list of references.

Comment 3: There are similar articles suc as Yu T, Guo J, Zhu S, Li M, Zhu Z, Cheng S, Wang S, Sun Y, Cong X. Protective effects of selenium-enriched peptides from Cardamine violifolia against high-fat diet induced obesity and its associated metabolic disorders in mice. RSC Adv. 2020 Aug 26;10(52):31411-31424. doi: 10.1039/d0ra04209a. PMID: 35520651; PMCID: PMC9056391.

Response 3: Thanks for the reviewer’s attention to our study. We cited this closely related article in Lines 74, 335, 340, 391, and 395. Though both Yu et al.’s and our studies used Cardamine violifolia (Cv) and got some similar findings, there were differences in the following main points:

  • Yu et al. collected selenium (Se)-enriched peptides with a Se content of 1177 ppm and a peptide content of 10% through the digestion of the protease compound (doi:10.1039/d0ra04209a). Their study focused more on the role of extracted Se-rich compounds in Cv.
  • In our study, we ground whole control Cv (Cv1) and Se-enriched Cv (Cv2) plants into powders and mixed them into diets with different fat contents to obtain Cv1D, Cv1HFD, and Cv2 Therefore, our study primarily focuses on the effects of whole Cv, as a hyperaccumulator of Se, on high-fat diet-induced obesity.
  • We also paid attention to the coenriched Cd in Se-rich Cv and called for further study on the potential risk of Cd in the product.

Reviewer 2 Report

Comments and Suggestions for Authors

The study conducted by Zhu et al. evaluated the effect of a Se-enriched diet on HFD-induced lipid disorder and liver damage. The study is original, unpublished and has evidence to be accepted for publication.

Why was the study conducted only with female mice?

Since the study involved animals, the authors must submit the completed SYRCLE form;

To reinforce the hypothesis of the study, as well as the results found, a toxicological analysis in similar groups would be interesting.

Author Response

Comment 1: Why was the study conducted only with female mice?

Response 1: We appreciate the reviewer’s attention to our study. The selection of female mice for this study is based on the following reasons:

  • Mice of the same species of different genders usually exhibit different susceptibilities to various treatments (doi:10.1038/s41366-018-0023-3; doi:10.1039/d1fo01241j), partly due to differences in endogenous estrogens, basal thermogenesis, physical activity levels, and mechanisms of energy expenditure (doi:10.1039/d1fo01241j; doi:10.3389/fnut.2022.828522). Sex steroids such as estrogen play a crucial role in female health and disease susceptibility, and gradually decrease with age as a natural biological process. There is evidence in animal studies that the reduction of estrogen exacerbated the adverse metabolic consequences of obesogenic diets (doi:10.3390/nu15061374). Meanwhile, given the apparent gender differences in the development of diet-induced obesity and metabolic disorders (doi:10.3389/fnut.2022.828522), it is worth noting that the prevalence rates of overweight and obesity remain constantly higher in female (doi:10.1016/j.metabol.2022.155217). Although selenium (Se)-enriched peptides from Cardamine violifolia (Cv) prevented high-fat diet (HFD)-induced obesity and metabolic disorders in male mice (doi:10.1039/d0ra04209a), there are no studies related to the effects of Cv on mitigating HFD-induced obesity in female mice.
  • Furthermore, as we mentioned in the manuscript, unlike Se-enriched extracts, both the control Cv (i.e., Cv1) and Se-enriched Cv (i.e., Cv2) could accumulate a certain amount of cadmium (Cd) during growth. There are reports that Cd affects health more commonly in females than in males (doi:10.1016/j.envres.2006.08.003; doi:10.1016/j.taap.2009.04.020), which may be due to the higher body accumulation of Cd in females. However, the risk of mortality is lower in females than in males in response to environmental Cd (doi:10.1289/ehp.11236), which may be attributed to estrogen protection in females.

Considering the reasons mentioned above, we used whole Cv plants and selected female mice as the subjects in our study.

Comment 2: Since the study involved animals, the authors must submit the completed SYRCLE form;

Response 2: Thanks for your suggestion, and we have finished the SYRCLE form. Please see the attachment.

Comment 3: To reinforce the hypothesis of the study, as well as the results found, a toxicological analysis in similar groups would be interesting.

Response 3: Thanks for your suggestion. In our manuscript, there are sections discussing Cd toxicity biomarkers for it was coenriched with Se in the plant. For instance, the detection of mRNA and protein levels of metal response element-binding transcription factor 1 (Mtf1) and metallothionein (Mt). They mainly participate in the detoxification of heavy metals, free radical scavenging, and protection against oxidative damage, and are highly sensitive to Cd exposure, being capable of inhibiting Cd toxicity (doi:10.1016/j.taap.2009.03.026; doi: 10.3390/ijms24010283). In Lines 414-436 of the manuscript, we discuss the observed changes in Mtf1 and Mt levels and their possible reasons. In fact, despite the accumulation of Cd in the mouse liver, we did not observe any clear evidence of Cd toxicity. We have compared the doses of Se and Cd with other studies involving the same species. In the intervention model, the Se/Cd ratio (1.5 ppm Se vs. 0.29 ppm Cd) in our study was 5.17, while in other studies showing the protective effect of Se against Cd exposure, the ratios were 0.16 (0.5 ppm Se vs. 3.2 ppm Cd) (doi:10.1016/j.jhazmat.2024.133812; doi:10.1016/j.ecoenv.2024.116101) and 0.8 (1.6 ppm Se vs. 2 ppm Cd) (doi:10.1016/j.bbagen.2018.04.009), much lower than ours. The naturally occurring low levels of Cd in Cv1 and Cv2 may not have reached the level required to induce toxic damage in the organism.